# Human-AI Shared Control via Policy Dissection

**Quanyi Li**[◇‡]**, Zhenghao Peng**[§]**, Haibin Wu**[◇]**, Lan Feng**[†]**, Bolei Zhou**[§]
[◇]Centre for Perceptual and Interactive Intelligence, [†]ETH Zurich,
[‡]University of Edinburgh, [§]University of California, Los Angeles

## Abstract

Human-AI shared control allows human to interact and collaborate with autonomous agents to accomplish control tasks in complex environments. Previous Reinforcement Learning (RL) methods attempted goal-conditioned designs to achieve human-controllable policies at the cost of redesigning the reward function and training paradigm. Inspired by the neuroscience approach to investigate the motor cortex in primates, we develop a simple yet effective frequency-based approach called *Policy Dissection* to align the intermediate representation of the learned neural controller with the kinematic attributes of the agent behavior. Without modifying the neural controller or retraining the model, the proposed approach can convert a given RL-trained policy into a human-controllable policy. We evaluate the proposed approach on many RL tasks such as autonomous driving and locomotion. The experiments show that human-AI shared control system achieved by *Policy Dissection* in driving task can substantially improve the performance and safety in unseen traffic scenes. With human in the inference loop, the locomotion robots also exhibit versatile controllable motion skills even though they are only trained to move forward. Our results suggest the promising direction of implementing human-AI shared autonomy through interpreting the learned representation of the autonomous agents. Code and demo videos are available at https://metadriverse.github.io/policydissect.

## 1 Introduction

In recent years, autonomous agents trained from deep Reinforcement Learning (RL) have achieved huge success in a range of applications from robot control [68, 56, 37, 41, 72], autonomous driving [25, 7, 29], to the power system in smart building [44]. Despite the capability of discovering feasible policy under unknown system dynamics in a model-free setting [66], the learning-based agents lack sufficient generalizability [14] in unseen environments, which hinders their real-world deployments. Furthermore, it remains difficult to understand the internal mechanism and the decision-making of the neural network [30, 78], especially when abnormal behaviors happen [10, 77, 49]. Thus the absence of generalizability, transparency and controllability limits the application of the end-to-end neural controllers in safety-critical systems.

One potential direction to make the neural controller safer and more trustworthy is incorporating human into the *inference* loop, which we refer to as *human-AI shared control*. With human involvement, the shared autonomy systems can achieve substantial improvement over the performance and safety [54, 15, 55], even on new tasks and unseen environments. Human-AI shared control has been implemented in various forms, such as enforcing intervention [28, 51] or providing high-level commands [15] to the AI models. Previous works mainly explore goal-conditioned reinforcement learning (GCRL) as a form of human-AI shared control, where a high-level goal is fed as an input to the policy during training [60, 2, 15, 25, 57]. Thus the behavior of the learned agent can be controlled by varying the input goal [47, 38, 79, 37]. However, training goal-conditioned policy requires additional modifications to the observation design, the reward function and the training

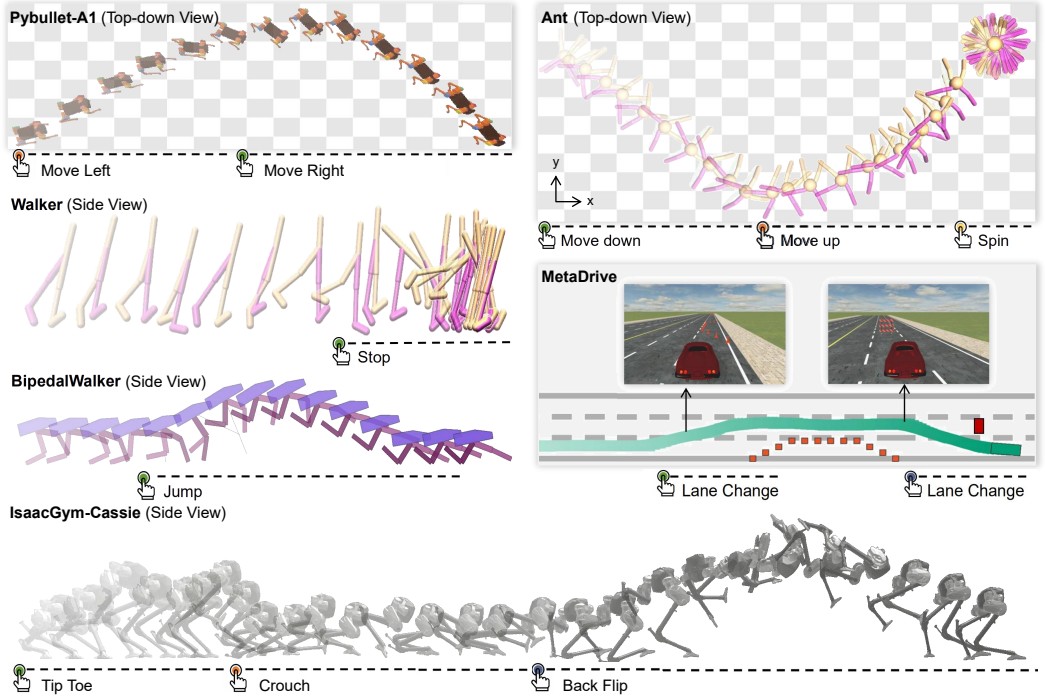

Figure 1: Examples of human-AI shared control enabled by *Policy Dissection*. Each autonomous agent is trained to solve its primal task without changing the environment like the reward function and observation. After training, the *stimulation-evoked map* is then generated by *Policy Dissection* as the interface for human to steer the agent. In various environments, behaviors such as jumping, spinning, redirection, backflipping can be triggered by human subject. These examples demonstrate that *Policy Dissection* is a general method for making a pretrained neural policy controllable.

process. In the case of bipedal robot for example, it is difficult to design reward functions for each behavior like crouching, backflipping, forward jumping. Also, initial state distribution and goal selection criteria should be meticulously designed for a successful GCRL training [50, 57]. On the other hand, the human control only happens at the goal level, still lacking in the interpretability of low-level neural policy.

In this work, we develop a minimalist approach called *Policy Dissection*, to enable human-AI shared control on autonomous agents trained on a wide range of tasks. *Policy Dissection* does not impose assumptions on the agent's training scheme like specially-designed reward function, and it requires neither retraining the controller's network nor modifying the environment. *Policy Dissection* achieves human-AI shared control by firstly dissecting the internal representations of the learned policy and aligning them with specific kinematic attributes, and then modulating the activation of the internal representations to elicit the desired motor skills.

Policy Dissection is inspired by the neuroscience studies on motor cortex of primates, where exerting electrical stimulation on different areas of motor cortex can elicit meaningful body movements [21]. Thus, a *stimulation-evoked map* can be built to reveal the relationship between the evoked body movements and the motor neurons located in different areas of motor cortex [21, 20, 34]. In order to replicate such a *stimulation-evoked map* inside a controller powered by artificial neural network, the proposed *Policy Dissection* conducts a frequency analysis and matching strategy for associating the kinematic attribute and the unit activation. The procedure of building a *stimulation-evoked map* by *Policy Dissection* is illustrated in Fig. 2. Concretely, we first roll out the given well-trained policy for many episodes and collect a set of time series of the activities of all units, and the variation in kinematics and dynamics. Afterwards, for each kinematic attribute we identify one unit whose activity align best with the variation of this kinematic attribute following the principle that there is the lowest frequency discrepancy between both the time series. After finishing the alignment for

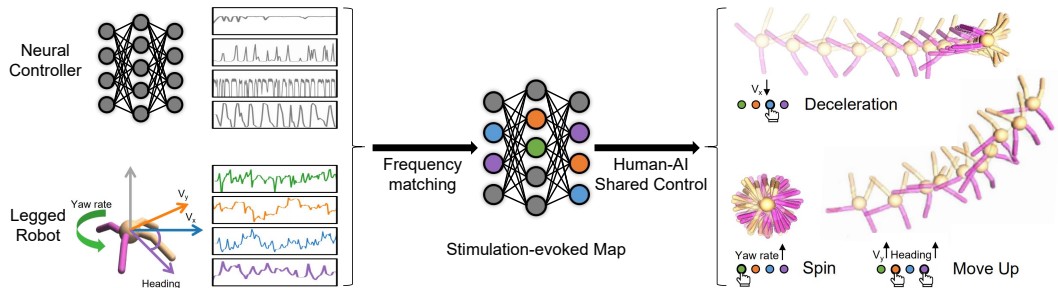

Figure 2: Overview of *Policy Dissection*. Taking the Ant robot as an example, our method first identifies the connection between *motor primitives* and kinematic attributes (painted in the same colors). After that, human can activate units to evoke desired behaviors by stimulating one or more *motor primitives*. For example, stimulating *motor primitive* related to yaw rate makes the Ant spin, while activating *motor primitive* related to velocities in different axes brings moving-up or deceleration.

all kinematic attributes, the identified units are then called *motor primitives*[1] whose activation can change corresponding kinematic attributes. Since a behavior can be described by the change of one or more kinematic attributes, the *stimulation-evoked map* can be built for recording the relationship of *motor-primitives*' activation and evoked behaviors. For example, the moving up behavior of the Gym-Ant shown in Fig. 2 can be described by turning upward (increasing yaw) and increasing upward speed ($v_y$), and thus we can activate corresponding two *motor primitives* to elicit the desired movement. In conclusion, *Policy Dissection* constructs an interpretable control interface on top of the trained agent for human to interact with and evoke certain behaviors of the agent, thus achieving human-AI shared control.

We evaluate the proposed method on several RL agents ranging from locomotion robots to autonomous driving agents in simulation environments. Experimental results suggest that meaningful *motor primitives* emerge in the internal representation. As shown in Fig. 1, novel behaviors can be deliberately evoked by activating units related to certain kinematic attributes, even though they are not the necessary behaviors to solve the primal task. In the quantitative evaluation, we use the human-AI shared control system enabled by *Policy Dissection* to improve the generalization in test-time unseen environment and achieve zero-shot task transfer. In an autonomous driving task, we first train baseline agents under mild traffic conditions and evaluate the driving policy on new test environments containing unseen near-accidental scenes. Different from the poor performance of the baseline agent in these new environments, the human-AI shared control system achieves superior performance and safety guarantee, with 95% success rate and almost zero cost. In the quadrupedal locomotion task, a robot with only proprioceptive state input can avoid obstacles with shared control, even though its primal task is moving forward. These results show that the proposed *Policy Dissection* not only helps understand the learned representations of the neural controller, but also brings the intriguing potential application of human-AI shared control.

## 2   Related Work

**Human-AI Shared Control**. Existing methods of human-AI shared control can be roughly divided into two categories. The first category is to have human in the training loop and then the well-trained policy can be executed without human in test time [80]. By incorporating human into the training, previous works successfully improve the performance on visual control tasks such as Atari game [1, 58, 70]. Robotic control tasks also benefit from human feedback [28, 43, 64, 51, 71]. The other category is to have human in both training and test time to accurately accomplish human-assistive tasks. A series of works study this problem based on the Atari Game [54, 59, 8, 31]. In addition, the human-AI shared control system is built on autonomous vehicle [19], robotic arms [27] and in multi-agent setting [31]. The reported results on various tasks indicate the effectiveness and efficiency of training and coordination with human-assistive AI. Unlike prior works, our method

---

[1]Motor primitive is used by researchers in biological motor control to indicate "building blocks of movement generation" [52, 32]

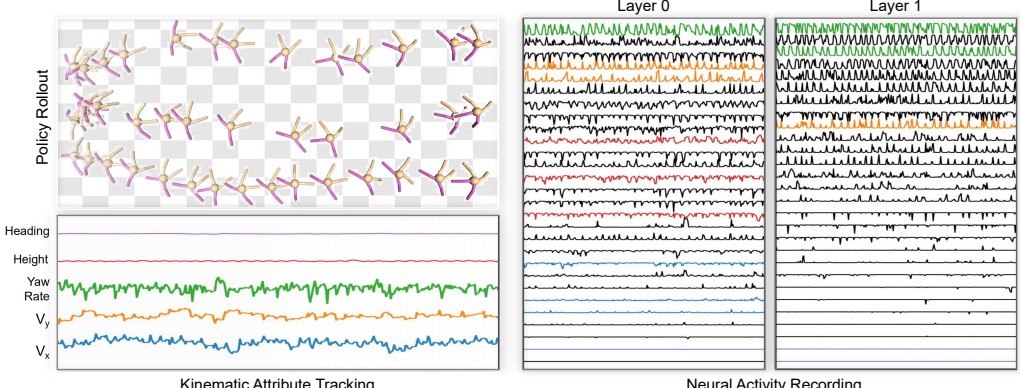

Figure 3: We first roll out the trained policy and record the neural activities and track kinematic attributes, like yaw and velocity. After frequency matching, kinematic attributes are associated with certain units, which are further called *motor primitives*. The curves of kinematic attributes and the aligned *motor primitive* are painted in the same colors. For clarity, we only show the result of one recorded episode and a proportion of units, and the curves of units are sorted by their amplitude.

allows human to collaborate with AI in test-time, while eschewing the need for any training-time human involvement. In experiments, shared control systems are implemented on several robotic control tasks to show the effectiveness of our method.

**Neural Network Interpretability**. There is a growing interest in the AI community to understand what is learned inside the neural network. Previous interpretability methods mostly focus on explaining the networks trained for visual tasks [81, 62, 11, 82, 40]. Various empirical studies have found that meaningful and interpretable structures emerge in the internal representation when trained for the primal tasks. For example, object detectors emerge in scene classifier [6], and disentangled layout and attribute controllers emerge from image generation model [74]. Varying the disentangled hidden features of the deep generative model can edit the content of the generated images. [5, 63, 73]. On the other hand, visuomotor control tasks can also acquire visual understanding to explain the behavior chosen by agent with visual input through saliency map [65, 3, 53, 12, 26, 22].

**Neuroscience Inspired Control**. The complex behaviors shown by recent autonomous agents prompt researchers to develop new analysis methods. As a source of inspiration, neuroscience provides useful tools [48, 34], and effective procedures [24, 18, 46] for understanding animal brains which are also like black-box systems. Recent works try to develop neural network-based policies from the perspective of neuroscience [17, 4, 76, 13]. In a closely related work on understanding learned agent, Merel et al. [46] use RSA [35] with CKA [33] similarity index to reveal that behaviors spanning multiple timescales are encoded in distinct neural layers. Apart from providing the neuroscientific tools, many concepts from neuroscience, such as *motor primitive* also inspire researchers to design generalizable controllers [69, 67, 45].

## 3 Policy Dissection Method

*Policy Dissection* aims at deriving a human-controllable policy from a well-trained RL agent. It has two unique features to enable human-AI shared control: First, it operates on a well-trained RL agent without re-training or modifying the original reward/environment and the policy architecture. We only need to roll out the agent for several episodes and log its neural activity and the change of kinematic attributes. Based on the collected data, kinematic attributes are aligned with certain units which are further termed as *motor primitives*. Thus our method is non-intrusive to the primal task and trained model, and applicable to a wide range of environments. Second, activating certain *motor primitives* can elicit generalizable motor skills that even the agent is not trained to perform. As shown in Fig. 1, the bipedal Cassie robot can be steered to tip toe, crouch and back flip even if it is only trained to move forward in the primal task. In the following subsections, we will introduce the four essential steps of the *Policy Dissection* method in detail.

### 3.1 Monitoring Neural Activity and Kinematics

Since multi-layer perceptron (MLP) is the most common network architecture used in many RL tasks, we assume the neural controller is an MLP with $\mathcal{Z}$ units, which has $L$ hidden layers and all layers have the same number of $I$ hidden units[2]. We denote a neuron by $l, i$ as $z^{l,i} \in \mathcal{Z}$, where $l = 1, ..., L$ is the index of layer and $i = 1, ..., I$ is the neuron index in each layer, and its output at time step $t$ is $z_t^{l,i}$. Let $J$ be the total number of kinematic attributes. The agent's kinematic attributes are represented by $\mathcal{S} = \{s^j\}$, $j = 1, ..., J$, and the measurement of each kinematic attribute $s^j$ at time step $t$ is $s_t^j$. We define the *motor primitive* as the unit $m^j \in Z$ that can steer the agent and cause a change of the corresponding kinematic attribute. We assume that each kinematic attribute $s^j$ has a corresponding *motor primitive* $m^j$ to drive the attribute to a desired state, and thus there are also $J$ *motor primitives*, denoted by $\mathcal{M} = \{m^j\}$, $j = 1, ..., J$. Our goal is to discover the *motor primitive* $m^j$ by associating the units with certain kinematic attribute $s^j$.

As shown in Fig. 3, we take the Ant robot as an example. *Policy Dissection* starts by rolling out the pretrained agent for several episodes and tracking the activation of all units as well as the kinematics, such as yaw rate and velocity. We record the $I \cdot L$ time series $\{\mathbf{z}^{l,i} = [z_1^{l,i}, z_2^{l,i}, ...]\}^{I \cdot L}$ measuring the neural activities and $J$ time series of kinematic attributes $\{\mathbf{s}^j = [s_1^j, s_2^j, ...]\}^J$ for further analysis.

### 3.2 Associating Units with Kinematic Attributes

Observing the neural activity and kinematic measurement recorded in Fig. 3, we can find that several repeated kinematic patterns appear in different frequencies when executing the policy, which is reminiscent of animal behavioral research [21, 20, 34]. It suggests that the motor skills has its predominant frequency component [46], controlling fast kinematics and slow kinematics. Since neural activation signals also encode distinct frequency information as shown in Fig. 3, this resemblance inspires us to make the following hypothesize:

*For one kinematic attribute, the unit with minimal frequency discrepancy is the motor primitive.*

To this end, we process time series of neural activity and kinematic attributes in frequency domain, where the difference of predominant frequency between two signals can be clearly revealed and the phase discrepancy of two time series are eliminated. As a consequence, this allows us to analyze the recorded data and associate kinematic attributes with units whose activity best matches the variation of target kinematic attributes. Concretely, all time series $\{\mathbf{x}\}$ including the neural activities $\{\mathbf{z}^{l,i}\}$ and change of kinematic attributes $\{\mathbf{s}^j\}$ are transformed into the frequency domain through Discrete-time *Short-time Fourier transform* (STFT):

$$\text{STFT}(d, \omega | \mathbf{x}) \equiv \sum_{t=-\infty}^{\infty} x_t \mathbf{W}(t - d) e^{-j\omega t}. \tag{1}$$

STFT splits the time series $\mathbf{x} = [x_1, x_2, ...]$ into a sequence of temporal window and $d \in [1, +\infty]$ denotes the time step of the center of the temporal window. STFT then conducts Fourier transform in each temporal window. $\omega \in [0, +\infty]$ denotes the frequency. To reduce spectral leakage and reserve the frequency information as much as possible, the window function $\mathbf{W}(t)$ used in *Policy Dissection* is *blackman window*. Moreover, the window length is $64$ with hop length $16$. We apply STFT instead of naive Fourier transform since it can retain temporal information by breaking the time series into several overlapping chunks of frames. This feature is important since in the following frequency matching we focus on the similarity of resulting spectrograms as well as the temporal correlation of two signals. We use the spectrogram to characterize a measurement:

$$\text{SG}(d, \omega | \mathbf{x}) \equiv |\text{STFT}(d, \omega | \mathbf{x})|^2. \tag{2}$$

After obtaining the spectrograms for all kinematic attributes and neural activities, *frequency discrepancy* can be computed for each neuron-kinematics pair $(\mathbf{z}^{l,i}, \mathbf{s}^j)$:

$$\text{Dis}(\mathbf{z}^{l,i}, \mathbf{s}^j) = \sum_d \sum_\omega \|\text{SG}(d, \omega | \mathbf{z}^{l,i}) - \text{SG}(d, \omega | \mathbf{s}^j)\|_2. \tag{3}$$

---

[2]To keep the notation concise, we assume all hidden layers of MLP have the same number of units $I$. MLPs with a varying number of hidden units per layer still come to the same conclusion.

For each kinematic attribute, *frequency matching* is applied to select the unit with minimal average discrepancy across $N$ collected episodes as the *motor primitive* responsible for triggering the change of desired kinematic attributes:

$$m^j = \underset{z^{l,i}}{\arg\min} \frac{1}{N} \sum_N \text{Dis}(\mathbf{z}^{l,i}, \mathbf{s}^j), \tag{4}$$

## 3.3 Building Stimulation-evoked Map

A behavior $b^k$ can be described by changing a subset of kinematic attributes $\mathcal{S}^k \subseteq \mathcal{S}$, which can be achieved by activating a set of corresponding *motor primitives* $\mathcal{M}^k \subseteq \mathcal{M}$. Therefore, these movement generation building blocks, *motor primitives*, are associated with certain behaviors, inducing the *stimulation-evoked map* $\{(\mathcal{M}^k, b^k)\}, k = 1, ..., K$, where $K$ is the number of discovered behaviors. Taking Cassie's back-flip shown in Fig. 1 as an example, this behavior can be described by increasing 1. height 2. pitch and 3. knee force. Therefore, we activate three corresponding *motor primitives* selected by Eq. 4 with output calculated by Eq. 8 to make the robot back flip. In practice, it is possible that more *motor primitives* are required. Again, taking the back-flip as an example, the force for both knee may be different after activating corresponding units, and thus giving the robot a rolling velocity, which can be further eliminated by activating roll related *motor primitive*.

## 3.4 Steering Agent via Stimulation-evoked Map

Now the last thing for evoking a behavior $b^k$ is to determine the output value for *motor primitives* $\mathcal{S}^k$. At time step $t$, activating the *motor primitive* $m^j$ related to kinematic attribute $s^j$ stands for applying its derivative $\dot{s}^j$ to the agent. For example, activating the speed related units equals to applying the derivative of speed, acceleration, to the agent. Therefore, changing the kinematic attribute $s^j$ at time step $t_0$ by activating the *motor primitive* $m^j$ with a unit output $v^j$ can be formulated as:

$$s_{t_1}^j = s_{t_0}^j + \int_{t_0}^{t_1} f(v^j)dt, \tag{5}$$

where $T = t_1 - t_0$ is the activation period of the *motor primitive* $m^j$, constant $v^j$ is the time-invariant output of $m^j$ and $f(v)$ is a mapping from *motor primitive* output value $v^j$ to $\dot{s}^j$.

For steering the agent, our goal is to understand some properties of $f(v)$, especially the correlation between $f(v)$ and $v$, so that we can amplify or reduce the $s^j$ by choosing a $v$ or $-v$. Taking the Ant robot of Fig. 1 as an example, activating the unit controlling y-axis movement with *positive* value can drive the Ant move left, and a *negative* value makes the Ant move right. We will illustrate the approach to probe the correlation in the following subsection.

**Identifying Correlation Coefficient**. We compute the correlation coefficient through *spectral phase discrepancy*. we first find the predominant frequency component for each signal:

$$\omega^* = \underset{\omega}{\arg\max} \sum_d \|\text{SG}(d, \omega | \mathbf{m}^j) - \text{SG}(d, \omega | \mathbf{s}^j)\|_2. \tag{6}$$

At each window of STFT, we can compute the phase discrepancy of the transforms of $\mathbf{m}^j$ and $\mathbf{s}^j$ at the predominant frequency $\omega^*$. The average phase discrepancy at all temporal windows is computed across $N$ collected episodes:

$$p^j = \frac{1}{D \cdot N} \sum_d \sum_N [\Phi(\text{STFT}(d, \omega^* | \mathbf{m}^j)) - \Phi(\text{STFT}(d, \omega^* | \mathbf{s}^j))], \tag{7}$$

where D is the number of temporal windows and $\Phi$ is the function retrieving the phase. We then normalize $p^j$ to get correlation coefficient $\rho$ between $m^j$ and $s^j$ as $\rho^j = 1 - |2p^j|/\pi$. $\rho^j$ tells us the most important thing that whether the $s^j$ will be amplified or reduced by overwriting the output of $m^j$ with a *positive* or *negative* value. After that, $v^j$ is finally determined by

$$v^j = c^j \rho^j, \tag{8}$$

where $c^j$ is a magnitude coefficient determining the scale of $v^j$ and thus $\dot{s}^j$. We usually determine the $c^j$ by observing the performance after trying evoking the desired behavior with different $c^j$, while it can also be determined by an external PID controller, which takes the error $\hat{s} - s_t^j$ between target state $\hat{s}$ and current state $s_t^j$ as input and outputs $c^j$ automatically.

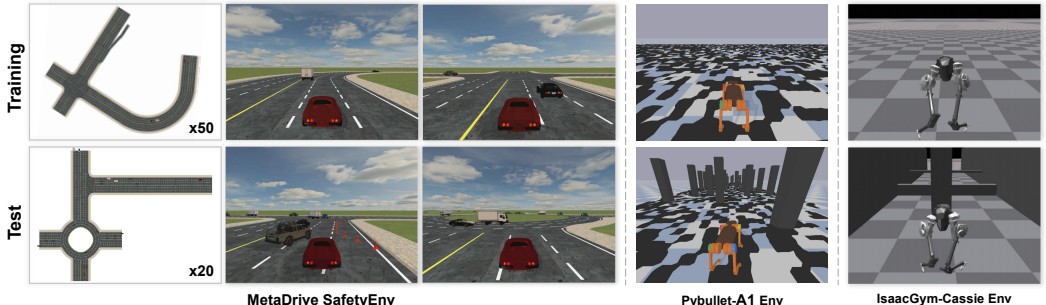

Figure 4: Examples of training and test scenes used in the generalization experiment. In MetaDrive SafetyEnv, the held-out 20 test scenes are more complex compared to 50 training scenarios. For quadrupedal robot locomotion task, the robot is trained to traverse open but bumpy terrains and tested on the terrain full of obstacles where obstacle avoidance ability is needed to reach the destination. In the Cassie environment, the agent is trained to move forward. However, crouching, jumping and sidestepping are required to reach the destination in the test environment.

## 4  Experiments

We first apply *Policy Dissection* to an autonomous driving task to study the learning dynamics in Sec. 4.2. We demonstrate that the emergent pivotal *motor primitives* are highly correlated with the success of policy learning. With the trained driving policy, we show a novel application of the shared autonomy in Sec. 4.3: improving test-time generalization, where Human-AI shared control can be an alternative to overcoming the generalization gap between training and unseen test-time environment. The experiment demonstrates that our method finds not only the correlation but causality between unit output and kinematics. In Sec. 4.4, we conduct experiments on quadrupedal robot and demonstrate that a quadrupedal robot trained to traverse complex terrain with only proprioceptive states can transfer to accomplish the obstacle avoidance task with a bit human effort, even if it does not have any information of the surrounding. For qualitatively evaluating our method, we provide a demonstration in Sec. 4.5, where a bipedal Cassie robot trained for moving forward can solve a challenging parkour environment with the help of human subject. In Sec. 4.6, we investigate the coarseness of the goal-conditioned control enabled by *Policy Dissection* through a command tracking experiment , and compare it with the state-of-the-art goal-conditioned controller in IsaacGym [42, 57].

### 4.1  Experimental Setting

The experiments conducted on Cassie and ANYmal in IsaacGym [57], Ant and Walker in Mu-JoCo [68] and the BipedalWalker [9] follow the general setting. Other experiments conducted on MetaDrive and Pybullet-A1 environments are introduced as follows:

**MetaDrive**. We train agents to accomplish autonomous driving task in the safety environment of MetaDrive [39]. In this task, the goal is to maneuver the target vehicle to its destination with low-level continuous control actions. An episode terminates only when the vehicle arrives at the destination, drives out of road, or collides with the sidewalk. The observation of autonomous vehicle consists of map, navigation, lidar, and state information such as speed, heading and side distance. More training environment details can be found in appendix.

**Pybullet-A1**. The legged robots locomotion experiments are conducted on the Pybullet-Unitree-A1 [16] robot. We follow the same environment setting used by Locotransformer robot [75], including terrain shape, obstacle distributions, sensors, reward definition, and termination condition. Two agents are trained. One "insensible" agent with only proprioceptive state is trained to walk on uneven terrain, while the other agent with Locotransformer is directly trained to avoid obstacles. The shared control system will be built on the "insensible" agent, where human subjects help the agent sidestep obstacles.

**Training Set and Test Set**. As shown in Fig. 4, for both autonomous driving and robot locomotion tasks, we prepare a training set and a held-out test set. For driving task, the test set is used to benchmark the test-time generalizability of human-AI shared control system and AI-only control system. We train autonomous driving policies in 50 different environments where the traffic condition

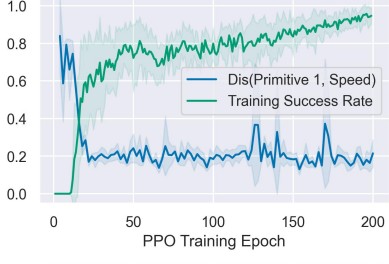

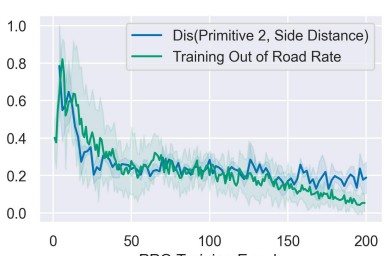

Figure 5: Learning Dynamics

| Method | Human Involvement | Episodic Return | Episodic Cost | Success Rate |
|---|---|---|---|---|
| Human | 405.57 ± 94.79 | 357.21 ± 38.3 | 0.47 ± 0.24 | 0.95 ± 0.03 |
| $SAC^{w/oH}$ | - | 263.85 ± 34.74 | 5.75 ± 0.25 | 0.75 ± 0.12 |
| $SAC^{H^n}$ | 26.52 ± 1.58 | 342.0 ± 6.25 | 0.42 ± 0.32 | 0.9 ± 0.0 |
| $SAC^{H^*}$ | 2.85 ± 0.22 | 353.19 ± 13.62 | 0.80 ± 0.14 | 0.95 ± 0.03 |
| $PPO^{w/oH}$ | - | 234.81 ± 23.45 | 3.37 ± 2.01 | 0.75 ± 0.03 |
| $PPO^{H^n}$ | 30.58 ± 0.18 | 352.99 ± 6.59 | 0.32 ± 0.12 | 0.92 ± 0.02 |
| $PPO^{H^*}$ | **2.55** ± 0.35 | **359.67** ± 11.20 | **0.05** ± 0.08 | **0.95** ± 0.02 |
| Relu-$PPO^{H^*}$ | 4.65 ± 0.3 | 352.98 ± 7.35 | 0.35 ± 0.1 | 0.88 ± 0.07 |
| Deep-$PPO^{H^*}$ | 4.88 ± 1.17 | 334.09 ± 2.91 | 0.48 ± 0.02 | 0.9 ± 0.02 |

Table 1: Improvements on the performance and the safety on test environments with human-AI shared control. $H$ indicates "human". $H^n$ and $H^*$ indicates two takeover methods. $H^n$ requires human to provide steering and throttle in the whole takeover period, while $H^*$ only asks to activate *motor primitives* at the start of takeover. We also do ablation study on the activation function and number of hidden layers of MLP. These results show our method is general and robust to the policy architecture and RL methods.

is mild (averagely 2 cars per map) and all traffic vehicles and the target vehicle can drive smoothly towards their destinations. No obstacles are in these environments. In testing time, the trained agents will be evaluated in the other 20 unseen maps with *higher* traffic density (averagely 6 cars per map). Besides, obstacles like traffic cones and breakdown vehicles will scatter randomly on the road. In this case, the target vehicle needs to be maneuvered delicately.We compare the performance and safety of the trained policy in two scenarios: runs in the test set solely, and runs with the human-AI shared control system built from *Policy Dissection*.

The primal task for the legged robot is to move toward one direction as fast as possible. We design the test environment to be a forest-like environment where tree obstacles are randomly scattered. Therefore, though the overall goal is still to move forward, the agent needs extra effort to sidestep obstacles in the test environment.

**Evaluation Metrics**. For driving task, we evaluate agents on all the 20 test environments, and report the ratio of episodes where the agent arrives at the destination as the *success rate*. Also, *Out of Road rate* can be calculated if the termination is caused by driving out of road. For benchmarking the safety of agents, the collision to vehicles, obstacles, sidewalk and buildings raises a cost $+1$. In addition, human attention cost is a huge concern of human-AI shared control system. We quantify this using *human involvement*, which is measured by the number of environment steps when the human subject presses keys for activating units, or controls throttle, brake, and steering wheel. We average the sum of reward, cost and human involvement on 20 environments as *episodic return*, *episodic cost* and *human involvement*. The above evaluation process will be repeated 5 times for each agent, and we report the average value and std for all metrics.

For the locomotion task, we use *moving distance* to measure the performance besides reward, since it directly reflects the goal. Moving distance is the displacement of robot on a specific direction, i.e. x-axis. The evaluation is also repeated for 5 times on the exclusive test environment.

**Human Interface.** We use keyboard as interface in all human-AI shared control experiments. When a specific key is pressed for evoking a behavior, related *motor primitives* will be activated and output $v^j$. The behavior termination can be triggered by ways: 1. Press another key and change to another behavior 2. The length of activation period $T = t_1 - t_0$ reaches a predefined length. Once a behavior is terminated, the related *motor primitives'* output will not be overwritten. In addition, there is one key mapping to the default behavior that no *motor primitives* are activated.

All experiments without human are repeated 5 times with different random seeds. $N = 5$ episodes are used for applying *Policy Dissection*. For experiments involving human subject, we repeat each experiment with 5 human subjects. For driving task, all human subjects have driving license. Human subject can pause the experiments if any discomfort happens. No human subjects were harmed in the experiments. Human subjects have signed the experiment agreement and get compensation.

## 4.2 Policy Dissection for Understanding Learning Dynamics

In this section, we verify the correlation between *motor primitives* and relevant kinematic attributes. We first train PPO [61] agent on the MetaDrive's training environments and then conduct *Policy Dissection* on the trained policy and find two *motor primitives* in which **Primitive 1** is associated with **Speed** and **Primitive 2** associated with **Side Distance**. We then plot the evolution of the frequency discrepancy between the identified *motor primitives* and the kinematics alongside with the macroscopic measure of the learning progress in Fig. 5. Note that the frequency discrepancies $\mathbf{d} = [d_1, ..., d_T]$ is normalized to $[0, 1]$ by $\tilde{\mathbf{d}} = \frac{\mathbf{d} - \min(\mathbf{d})}{\max(\mathbf{d}) - \min(\mathbf{d})}$.

The upper plot in Fig. 5 shows Primitive 1 gradually emerges to control speed, indicated by the decreasing discrepancy. With strong negative correlation, the success rate rapidly increases when the discrepancy of speed primitive also rapidly decreases between epoch 20-30. Similarly, in the lower plot in Fig. 5, the decreasing discrepancy between Primitive 2 and side distance implies that the internal unit of the agent is gradually specialized to control side distance. There exists a strong correlation between the discrepancy and the out-of-road rate, which can only be reduced by improving the ability to keep side distance. We thus conclude that the pivotal primitives emerge during training.

## 4.3 Human-AI Shared Control for Test-time Generalization

Using the *motor primitives* identified above, we can determine the activation value of *motor primitive* for steering associated kinematics according to Eq. 8. We apply the same process to dissect the policy trained from SAC [23] with the same network structure and build a *stimulation-evoked map* for shared control. We then invite human subject to collaborate with the SAC and PPO agents on the test environments with different takeover methods. As shown in Tab. 1, the average human involvement, episodic cost, and success rate in the test environments are reported across policies with or without human involvement. Human involvement greatly improves performance and safety when deploying the policy in unseen environments where delicate maneuvers such as side-passing and yielding are evoked by human, if necessary, to help finish the task.

Different from the original PPO policy which is an MLP with 2 hidden layers, 256 units per layer and "tanh" activation function, we also repeat the same experiment for Deep-PPO agent which has 6 hidden layers, and Relu-PPO which has "Relu" as activation function. The results reported in 1 show our method is invariant to the network architectures and RL algorithms.

## 4.4 Human-AI Shared Control for Task Transfer

In addition to the improved test-time generalization, the *stimulation-evoked map* enables RL agent to accomplish new tasks with the help of human. Recent quadrupedal robots trained by RL exhibit promising ability to walk on bumpy terrain via end-to-end control [36]. However, the agent only takes proprioceptive states as input without the observation of surroundings, and thus it can not avoid obstacles on

| Method | Reward | Moving Distance |
|---|---|---|
| State$^{w/oH}$ | $18.23 \pm 10.55$ | $6.55 \pm 0.04$ |
| State$^{H^*}$ | $413.93 \pm 93.33$ | $16.94 \pm 3.08$ |
| State+Vision | $445.47 \pm 190.85$ | $18.64 \pm 6.45$ |

Table 2: Test on Obstacle Avoidance Task.

its own. As shown in Fig. 4, we train the agent only to walk on uneven terrain but test it in a new environment full of obstacles. By applying our method, *motor primitives* related to yaw rate and speed control can be found and serve as the interface for human to steer the "insensible" quadrupedal robot to avoid obstacles. This forms the "State$^{w/oH}$" method. We also include "State+Vision" method [75], which trains a visuomotor policy for legged robot directly in the test environment, enabling it to sidestep obstacles. As shown in Table 2, with human involvement, *Policy Dissection* successfully transforms the "insensible" policy that is not fit to this task and greatly improves the performance, achieving comparable performance to the agent specially designed for this task.

## 4.5 Qualitative Result: Parkour Demo

As shown in the Fig. 4, the training objective of the bipedal Cassie robot is to move forward. After training, *Policy Dissection* can discover several skills in the learned representation, as shown in Fig. 6. Furthermore, these skills can be triggered and controlled by human subject for solving the challenging parkour environment. See the demo video for detail.

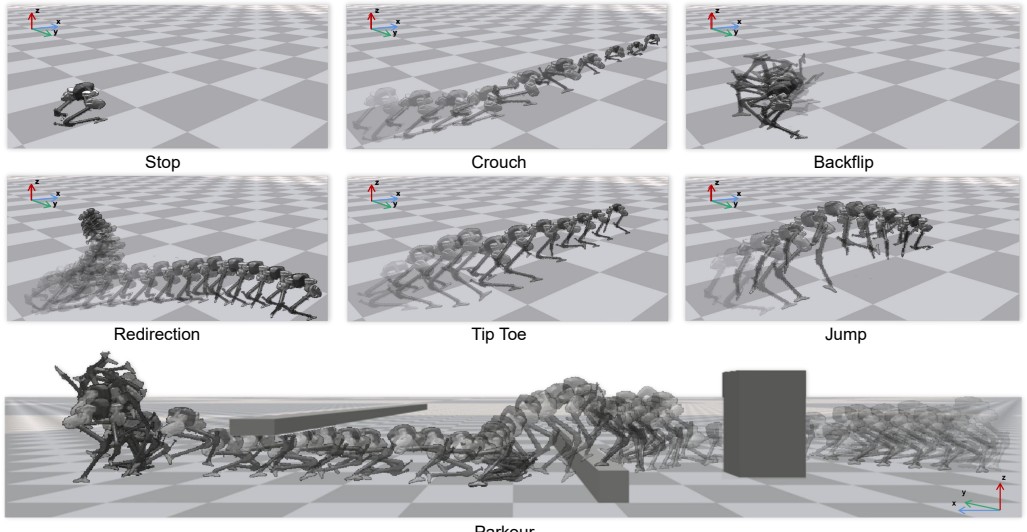

| Stop | Crouch | Backflip |
| Redirection | Tip Toe | Jump |

Parkour

Figure 6: *Policy Dissection* can discover several skills such as crouching, jumping, backflipping in the learned representation of the bipedal Cassie robot, which is originally trained to move forward. These skills can be combined by human subjects and overcome a challenging parkour environment.

## 4.6 Experiment on the Control Precision

We conduct an experiment to evaluate the control precision enabled by our method. We train a goal-conditioned policy for ANYmal-C robot in IsaacGym [57]. The trained robot taking raw rate as input can track a target yaw with a PID controller, and thus serves as the baseline. For a fair comparison, we directly dissect this policy and identify the yaw primitive. Likewise, a PID controller that takes the yaw error as input is used to automatically determine the output $v^j$ for yaw primitive, enabling tracking target yaw. Note that for the primitive activation control the explicit heading command in input is set to 0. Heading tracking results in Fig. 7 show that the control achieved by primitive activation has comparable precision to the goal-conditioned control (see demo video for more detail).

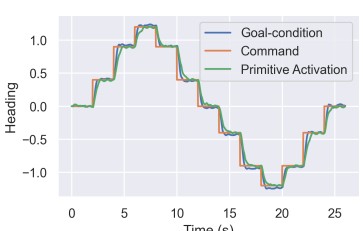

Figure 7: Heading tracking error from the goal-condition control and the primitive activation control.

## 5 Conclusion and Discussion

Inspired by the neuroscience approach to study the motor cortex in primates, we present a simple yet effective method called *Policy Dissection* to align internal neural representations and kinematic attributes of a pretrained neural policy. The resulting *stimulation-evoked map* allows human to steer the agent and evoke its certain behaviors. *Policy Dissection* enables human-AI shared control systems on top of the models trained from standard RL algorithms, without any re-training or modification. We quantitatively evaluate the human-AI shared control system in autonomous driving and locomotion tasks, and the result suggests the substantial improvement of the test-time generalization for task transfer. We further provide interactive demonstrations in various environments to show the generality of our method and its application for human-AI collaboration.

**Limitations and societal impact**. One limitation of *Policy Dissection* is that we only evaluate this method in continuous control tasks with the state vector as input. A wider range of tasks and neural network architecture such as discrete control tasks and CNNs can be adopted in future research. One potential negative social impact is that the safety issue remains to be solved when deploying shared control systems in real-world applications. As shown in the autonomous driving experiment, our method derived from empirical study has no theoretical guarantee on the safety of human subjects. The unsuccessful collaboration or conflict between human and AI may lead to potential risks.

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
