# OpenReview forum: "Human-AI Shared Control via Policy Dissection"
_NeurIPS.cc/2022/Conference — NeurIPS 2022 Accept_

### Official Review · Reviewer_cucK · 2022-07-06

**Rating:** 7
**Confidence:** 4
**Soundness:** 3 good
**Presentation:** 2 fair
**Contribution:** 3 good

**Summary:**

This paper introduces a method termed 'policy dissection', which finds a mapping between neurons in a trained continuous control policy and high-level kinematic motions of interest (eg. turn left, right, stop, go forward). This mapping is then used to allow for human intervention of modification of the trained policy. The core contributions are a method for identifying this mapping (using a frequency-based analysis technique), and to show that this can be exploited to allow policy intervention (by amplifying units in a direction that correlates with an increase in the desired kinematic motion) and high level policy modification by humans.

**Questions:**

I have a fair idea of how the mapping between neurons and kinematic primitives of interest is identified, but I am less clear about how the reverse is implemented, and how a user keypress is turned into a unit amplification. Please could you explain this in more detail? Eg. How much do you amplify, is this temporary and lasting only a few seconds, or a permanent change.

I would value hearing whether the perceived coarseness of the intervention control is due to the relatively naive mapping (only really rough correlations) or whether this could be improved through a better human intervention control policy?

Minor comments:

This work reminded me of control loop pairing work (relative gain array) where the best input output pairings are selected for multivariate control systems. I would value hearing your perspective on future work and what is needed to get finer control over motion primitives from the network? Does this need better mappings, or better eg. hierarchical policies?

Can you be more specific about the metadrive tasks used to populate Table 1, and the different takeover methods.

**Limitations:**

Limitations are discussed (neurons that do not align with any primitives), but I would value more discussion on the coarseness of the method - the video does make human take-over appear quite crude. I would also value a discussion of competing approaches, eg. policies that are explicitly trained to allow for kinematic intervention, and motivation for the proposed approach over this.

**Strengths And Weaknesses:**

Strengths:

- This is a creative and interesting idea, which I have not come across before. I enjoyed reading this paper.
- I think this work is significant from the perspective of trying to understand the implicit representations or primitives captured by neural control policies, and because it introduces a method for humans to controllably modify a learned policy to achieve a goal.
- A thorough set of experiments are provided on a number of experimental settings.

Weaknesses:

- The paper could do with some smoothing, and improvements to the writing. There are numerous typos and grammatical errors, but these did not get in the way of the idea being communicated.
- Although the proposed approach does allow for some level of human intervention and modification, this appears to be very imprecise and it is unlikely that the proposed approach allows more than very coarse control.

---

> ### Author Response · Authors · 2022-08-02
> **Response to Reviewer cucK**
>
> Thank you very much for providing such a detailed review and suggestions for improving the clarity and quality of our paper! Apart from the paper revision and typo fixing, we have the following responses for your questions.
>
> ---
> Q1: Although the proposed approach does allow for some level of human intervention and modification, this appears to be very imprecise and it is unlikely that the proposed approach allows more than very coarse control. I would value hearing whether the perceived coarseness of the intervention control is due to the relatively naive mapping (only really rough correlations) or whether this could be improved through a better human intervention control policy.
>
>
> A1: We designed a new yaw tracking experiment to evaluate the control precision. It shows that our method can achieve comparable control accuracy, compared to goal-conditioned control on the Anymal-C robot.
> Also, we equip the motor primitive with a PID controller to automatically determine the output of the yaw velocity primitive. It allows the dissected policy to track a yaw command, which showcases another human-intervention control policy, issuing a target command. Please find more details in the common response.
>
> ---
> Q2: I have a fair idea of how the mapping between neurons and kinematic primitives of interest is identified, but I am less clear about how the reverse is implemented, and how a user keypress is turned into a unit amplification. Please could you explain this in more detail?
>
> A2: When a certain key is pressed, the output of related motor primitives will be overwritten with $v^j$ according to Eq. 8 until another key is pressed,  which will trigger new motor primitives and then recover the activation of previous ones. The change of corresponding kinematics will be determined by the length of the activation period. For example, if we want to decrease the speed of an autonomous vehicle by activating speed primitive, the longer the activation time, the larger the speed will be decreased. Besides, there is always one key mapping to none of the motor primitives. Pressing this key will recover the default policy, where no intervention is applied. We have added more descriptions of the experiment setting in Section 3 in the revision.
>
>
> ---
> Q3: This work reminded me of control loop pairing work (relative gain array) where the best input-output pairings are selected for multivariate control systems. I would value hearing your perspective on future work and what is needed to get finer control over motion primitives from the network. Does this need better mappings, or better eg. hierarchical policies?
>
> A3: The experiment data collection and analysis process can be associated with some classical control methods, like RGA and DMD [1][2]. These methods are equipped with theoretical guarantees. Therefore, one important future direction is to probe the internal machenism of the neural controllers with classical control tools.
>
> ---
> Q4: Can you be more specific about the MetaDrive tasks used to populate Table 1, and the different takeover methods?
>
> A4: MetaDrive is a driving simulator that can generate countless driving scenarios whose difficulty can be controlled by changing the density and distribution of traffic flow and obstacles. We first train an autonomous driving policy via RL on 50 training maps where obstacles do not exist and traffic density is low (averagely 2 vehicles per map). After training, these agents are evaluated on 20 new test maps with different road structures. In addition, the average number of traffic vehicles is increased to 6, and obstacles like cones and barriers are randomly scattered on the map. In this situation, frequent lane changing and braking is required to sidestep obstacles or merge into traffic flows.
>
> As shown in Table 1, directly deploying the trained policy on these unseen difficult maps yields a high cost and low success rate. One naive way to improve the performance is to introduce humans into the control loop and allow takeover. At each timestep, if specific keys (w, a, s, d) are pressed, the corresponding control signal like throttle/steering will be overwritten, just like playing a racing game with a keyboard. This takeover paradigm requires human control during the whole takeover period, which is costly. However, the shared-control system built by our method only needs to press once to trigger corresponding motor primitives. For example, by pressing the “braking” key once, the speed of the vehicle will be decreased to 0 automatically, while the naive method needs to hold the “braking” key until the vehicle is static.
>
> ---
> Reference:
>
> [1] Schmid, P. J. (2010). Dynamic mode decomposition of numerical and experimental data. Journal of fluid mechanics, 656, 5-28.
>
> [2] Proctor, J. L., Brunton, S. L., & Kutz, J. N. (2016). Dynamic mode decomposition with control. SIAM Journal on Applied Dynamical Systems, 15(1), 142-161.

---

> > ### Comment · Reviewer_cucK · 2022-08-04
> > **Thanks**
> >
> > Thank you for your response to my questions

---

> ### Author Response · Authors · 2022-08-02
> **Response to Reviewer cucK**
>
> Q5: I would value more discussion on the coarseness of the method - the video does make human take-over appear quite crude. I would also value a discussion of competing approaches, eg. policies that are explicitly trained to allow for kinematic intervention, and motivation for the proposed approach over this.
>
> A5: The yaw tracking experiment in common response shows that our method also allows precise control. Actually, the main goal of this work is to provide a new frequency-based perspective on the internal mechanism of the neural controller and make this black-box system more transparent. The most effective way to verify the correlation and causality identified by policy dissection is to conduct shared-control experiments where human can activate specific units to change the kinematics. We also provide a new bipedal walker demo in common response. It shows diverse skills like back-flipping, forward jumping and crouching in a parkour setting. Hope this work can inspire more future works toward interpretable neural decision-making and human-AI shared control applications.

---

### Official Review · Reviewer_XzPE · 2022-07-11

**Rating:** 4
**Confidence:** 2
**Soundness:** 2 fair
**Presentation:** 1 poor
**Contribution:** 2 fair

**Summary:**

  This paper proposes a framework where a human can intervene to steer the behaviour of an RL agent. This is done by allowing the human to select from a "stimulation-evoked map".

**Questions:**

* How are the motor primitives to be stimulated found/labelled? For example, how is the spinning primitive added? Are these predefined or learnt?

**Limitations:**

Agent isn't able to learn from interventions

**Strengths And Weaknesses:**

* I think the paper has a big problem with clarity, the setting is very unclear.

* Experiments don't compare against a baseline

* Requires full attention from the human and system isn't able to learn from the human interventions



Apologies for not discussing this more but this is not my area and I found the paper really unclear, so I struggle to write a proper review.

---

> ### Author Response · Authors · 2022-08-02
> **Response to Reviewer XzPE**
>
> Thank you for the comments and the acknowledgment of low confidence. It is possible that the bidding system assigns some less related papers to reviewers. Despite this, your review is still meaningful. We already added more discussion to the related work section about the difference between our shared-control setting and the traditional learning-from-intervention setting.
>
>
>
> Previous methods acquire a shared-control policy by having human in the training loop [1][2] and minimizing training time human intervention, which have a different setting to our method. Instead, the shared control’s setting is to have human in the inference loop. Thus the proposed Policy Dissection directly identifies controllable skills by running the well-trained policy and analyzing the collected data on the frequency domain. These skills, furthermore, can be triggered and combined randomly by human or a high-level controller to achieve shared-control or hierarchical control in testing time, without retraining the agent model. We provide a new bipedal Cassie robot video to help understand our method. Please find it at https://drive.google.com/file/d/1k182aDoS7SyddeO4Gu_ls_FHeXwkLBeh/view?usp=sharing
>
> ---
> Reference:
>
> [1] Spencer, J., Choudhury, S., Barnes, M., Schmittle, M., Chiang, M., Ramadge, P., & Srinivasa, S. (2020). Learning from interventions: Human-robot interaction as both explicit and implicit feedback. In 16th Robotics: Science and Systems, RSS 2020. MIT Press Journals.
>
> [2] Kelly, M., Sidrane, C., Driggs-Campbell, K., & Kochenderfer, M. J. (2019, May). HG-DAgger: Interactive imitation learning with human experts. In 2019 International Conference on Robotics and Automation (ICRA) (pp. 8077-8083). IEEE.
>
> [3] Saunders, W., Sastry, G., Stuhlmueller, A., & Evans, O. (2017). Trial without error: Towards safe reinforcement learning via human intervention. arXiv preprint arXiv:1707.0517

---

> ### Author Response · Authors · 2022-08-06
> **Any post-rebuttal suggestions?**
>
> Dear Reviewer,
>
> Please kindly let us know if you have any post-rebuttal feedback. Besides illustrating the difference from previous human-in-the-loop works, it will be great to see the new video can make the method and its application clear. Look forwards to hearing back about your optimistic acknowledgment of this submission.
>
> Best,
> Authors

---

> ### Author Response · Authors · 2022-08-09
> **Have the response addressed the reviewer's concerns?**
>
> Dear Reviewer,
>
> Thank you for the initial review. We prepared a detailed response below. Could you please provide some feedback and reassess our paper? If something is still unclear, we would be happy to provide alternative explanations or clarifications on any other part of the paper.
>
> Thanks,
> Authors

---

> ### Comment · Reviewer_XzPE · 2022-08-09
> **response**
>
> I'm increasing my score to borderline reject. I can't go over that because I still feel that the paper should be must more understandable, even for people that are not in the area. I feel there's space to improve the writing.

---

### Official Review · Reviewer_m4BT · 2022-07-12

**Rating:** 4
**Confidence:** 3
**Soundness:** 2 fair
**Presentation:** 3 good
**Contribution:** 3 good

**Summary:**

In this paper, the authors propose a novel method to modify the behavior of pretrained neural network policies to obtain the desired control effects without changing the weights of the networks unchanged. Instead, the authors analyze the activation patterns of all neuron units in the network, and create associations between the neuron activation and desired kinematic properties (such as angular velocity, translational velocity, etc) in the frequency domain through Fourier transforms. For each kinematic attribute, the proposed method identifies one evoking neuron; So during the online policy roll out the users can directly stimulate a certain desired behavior of the robot by activating the corresponding neuron, even though the policy is never trained for this new behavior/task. The authors demonstrate in a set of experiments that, with their method and human-in-the-loop, they can improve the performance of many safety environments.

**Questions:**

Other questions and suggestions:

The Figure 6 is a bit confusing. It does not make sense to have discrepancy distance and training success rate curves share the same y-axis.

**Strengths And Weaknesses:**

The strengths of the paper:

It introduces a new concept to do zero-shot generalization of existing policies without any re-training.

Provides a new insight on the interpretability of neural network policies, i.e. from the frequency domain to associate neuron activations with behavior.

Demonstrates that, even though a policy may not be trained with certain tasks, some behaviors might have already been seen during the training phases and somehow are “memorized” through the network structure.

The weaknesses of the paper:

The motivation of the paper is to make neural controllers safer by incorporating humans in the loop. Thus, I assume that control accuracy and robustness of the NN policy is very important. However, by “hacking” into a NN policy that is not trained with human inputs (e.g. steering, throttle cars), I am not convinced if the behavior, such as velocity tracking accuracy, etc,  can be as good as goal/human input conditioned policies.

The authors assume that for each kinematic attribute, there is one neuron that is responsible for. This assumption is not quite sound to me, as the activation of a neuron will have chain effects for other neurons down the propagation path. And the other neurons might be responsible for other kinematic behaviors according to the same assumption.

Also, I am not sure how this method will scale with truly deep neural networks with many hidden layers and neurons, as there are many more activation paths for the network.

---

> ### Author Response · Authors · 2022-08-02
> **Response to Reviewer m4BT**
>
> Thank you for the detailed review! We appreciate your insightful questions which improve this submission. Please see the responses below.
>
> ---
>
> Q1: The motivation of the paper is to make neural controllers safer by incorporating humans in the loop. Thus, I assume that the control accuracy and robustness of the NN policy are very important. However, by “hacking” into a NN policy that is not trained with human inputs (e.g. steering, throttle cars), I am not convinced if the behavior, such as velocity tracking accuracy, etc, can be as good as goal/human input conditioned policies.
>
> A1: We have conducted yaw tracking experiments that compare the control accuracy of goal-conditioned control and primitive activation control (ours). It shows that our method achieves similar command tracking accuracy, compared to the state-of-the-art goal-conditioned controller trained for Anymal-C robot in IsaacGym. Please find the experiment with a video in the common response.
>
> ---
> Q2: The authors assume that for each kinematic attribute, there is one neuron that is responsible for it. This assumption is not quite sound to me, as the activation of a neuron will have chain effects on other neurons down the propagation path. And the other neurons might be responsible for other kinematic behaviors according to the same assumption.
>
> A2: Good question. The chain effect is exactly the reason why the shared-control system enabled by policy dissection works. Take the quadrupedal robot as an example. If we want to change its heading, we have to steer its joint position. When activating the heading primitive, it is the chain effect that triggers other primitives responsible for joint position control. In the future, we will try probing the hierarchical distribution of motor primitives in the neural controllers to investigate the chain effect. Furthermore, we don’t assume one neuron is responsible to exact one attribute. Instead, a group of units can be identified to correspondto one attribute while one unit might contribute to multiple related attributes.
>
> ---
> Q3: I am not sure how this method will scale with truly deep neural networks with many hidden layers and neurons, as there are many more activation paths for the network.
>
> A3: MLP is actually the most commonly used network architecture for robotic control tasks in the real world [1][2][3]. The number of neural layers usually ranges from 2 to 6. In the experiment section, we already conducted shared control experiments on NN with 2 and 6 hidden layers in Table 1 and demonstrate that our method still works even if the number of the hidden layer is up to 6.
>
> ---
> Q4: Figure 6 is a bit confusing. It does not make sense to have discrepancy distance and training success rate curves share the same y-axis.
>
> A4: Thanks for pointing out this. The discrepancies $\mathbf{d}=[d_1,...,d_T]$is normalized with equation: $\frac{\mathbf{d} -min(\mathbf{d})}{max(\mathbf{d})-min(\mathbf{d})}$, so that they are both in range [0, 1] and can share the same y-axis.
>
> ---
> Reference:
>
> [1] Hwangbo, J., Lee, J., Dosovitskiy, A., Bellicoso, D., Tsounis, V., Koltun, V., & Hutter, M. (2019). Learning agile and dynamic motor skills for legged robots. Science Robotics, 4(26), eaau5872.
>
> [2] Wu, P., Escontrela, A., Hafner, D., Goldberg, K., & Abbeel, P. (2022). DayDreamer: World Models for Physical Robot Learning. arXiv preprint arXiv:2206.14176.
>
> [3] Li, Z., Cheng, X., Peng, X. B., Abbeel, P., Levine, S., Berseth, G., & Sreenath, K. (2021, May). Reinforcement learning for robust parameterized locomotion control of bipedal robots. In 2021 IEEE International Conference on Robotics and Automation (ICRA) (pp. 2811-2817). IEEE.

---

> ### Author Response · Authors · 2022-08-06
> **Any more suggestions or comments?**
>
> Dear Reviewer,
>
> Please kindly let us know if you have any post-rebuttal feedback. It will be great to see our rebuttal can address your concerns to some degree and you would become more optimistic about our submission along with all the other positive reviewers.
>
> Best,
> Authors

---

> ### Author Response · Authors · 2022-08-09
> **Have the response addressed the reviewer's concerns?**
>
> Dear Reviewer,
>
> Thank you for raising a number of questions in the initial review. We prepared a detailed response below. Could you please provide some feedback and reassess our paper? If something is still unclear, we would be happy to provide alternative explanations or clarifications on any other part of the paper.
>
> Thanks,
> Authors

---

### Official Review · Reviewer_g9wy · 2022-07-13

**Rating:** 6
**Confidence:** 4
**Soundness:** 2 fair
**Presentation:** 3 good
**Contribution:** 3 good

**Summary:**

The paper proposes a method to dissect a policy trained for a simple task to extract "motor primitives" (or set of neurons) that provides different behaviors or attributes. The authors then use these attributes to have humans control the agent for more complex tasks or to create different behaviors by combination. The paper showcases the algorithm in 2 domains: Driving and quadruped locomotion (in simulation). The authors first train for driving (or walking) forward, without any explicit goal conditioning. Then, the authors dissect the policy for different behaviors by replaying trained policy and doing frequency matching. Next, humans use the provided interface for obstacle avoidance. In addition, the authors show that they can obtain very different behaviors by combining these motor primitives in different domains such as 2D walker. As an example, they obtain front flipping behavior by activating jumping and roll rate which are discovered using policy dissection.

**Questions:**

Please see the weaknesses mentioned above. Could you address if it is possible to implement the suggested directions above? Otherwise, why not?

**Limitations:**

The limitations of the method is explained well, but does not contain the weaknesses (or potential directions) that I explained above. I don't think that the paper contains any major societal impact.

**Strengths And Weaknesses:**

The method proposed by the authors is very interesting for interpretability and discovery of skills in Reinforcement Learning. As the authors state, instead of training for complex behaviors, the authors can train for simple ones and discover different skills from replay. I think the front flipping is a very good example. It is very hard to design reward functions for all these different skills and train a network that can perform all of them, but the authors can find these skills and combine them without reward engineering which is very interesting, especially from locomotion perspective where the training are usually goal conditioned not skill-rewarded.

Another interesting side-effect of the approach is from robotics perspective. Although the authors are working on simulation, not real robots, the proposed method allows discovery of skills with small amount of replay data. This can be a very interesting approach to the well known sim-to-real problem, where trained policies are not behave the same way in the real world. The proposed method can be used to discover skills on the hardware although it does not match the simulated behavior, this would lead to motor skills that work on the real robot.

In my opionion, the apprach has some weaknesses as well.
1- The authors use the approach to have humans control the robot to solve slightly more complex task. On the other hand, the obstacle avoidance problem can already be solved with goal conditioned policies. So the results does not fully exploit the method to its full capacity. Instead, the authors could try to achieve a problem that can't be solved with simple training. One example I can think of is to have a parkour setup that requires jumping, walking low.

2-The authors could automate the higher level controls instead of collaborating with humans. The proposed schema is perfect for a hierarchical reinforcement learning approach. Once skills are discovered, the authors could retrain with the higher level controls provided to the humans with additional sensory information that did not exist in the initial training. This could very well train in very short time and handle the complex scenarios.

3-The authors could autmoate the discovery of the skills. The literature contains many examples of such approaches based on maximizing entropy during training. Instead of using human designed motor primitives, learning could provide more diverse behaviors as well.

4-The discovery is based on replay of the very simple behavior. If I understand correctly, the matching is based on behaviors seen in very small timeframes (i.e. it turns slightly left and then slightly right to go straight). But, if the initially trained policy was great, the behavior could very well just go straight which would lead to failure to match (again, if I understand the matching process correctly). Adding disturbances (at the neuron level) to the replay would provide much more than the initial data set where the agent is just going straight. Of course this might require some more work and careful design of the disturbing process.

Despite these weaknesses, I think the method has a very good potential and very interesting to read and the provided experiments is considerably enough to show some of this potential as well.

---

> ### Author Response · Authors · 2022-08-02
> **Response to Reviewer g9wy**
>
> Thank you for your insightful review. According to your suggestions, we have conducted additional experiments to show more applications of the proposed method. Please find the detailed responses below.
>
> ---
>
> Q1: The results do not fully exploit the method to its full capacity. Instead, the authors could try to achieve a problem that can't be solved with simple training. One example I can think of is to have a parkour setup that requires jumping and walking low.
>
> A1: Thanks for your suggestion! Please find the new parkour demo in the common response.
>
>
> ---
>
> Q2: Authors could retrain with the higher level controls provided to the humans with additional sensory information that did not exist in the initial training. This could very well train in a very short time and handle complex scenarios.
>
> A2: We trained a high-level DQN agent in MetaDrive to make lane-changing/braking decisions by activating the identified motor primitives. With 20,000 steps, it adapts to the test environment with a 0.94 success rate. However, the high test-time cost indicates it is not safe. It is an appealing future direction to investigate how to do fast adaptation safely for RL agents.
>
> ---
>
> Q3: The authors could automate the discovery of the skills with mentioned related works
>
> A3: Sure. For example, the intrinsic reward based on mutual information [1][2][3] can be introduced to enrich the skill set. This will be left for future work, and this paper aims at illustrating a general policy dissection method for any native RL policies.
>
> ---
>
> Q4:  If the initially trained policy was great, the behavior could very well just go straight which would lead to failure to match.
>
> A4: Under this extreme situation, policy dissection indeed can not identify any meaningful primitives, since our method probes correlations from variation. However, as we said before, it is a rare or even impossible situation where the MDP achieves steady-state so that $|s_t|=|s_{t+1}|$. The main reason is that control tasks nowadays have huge continuous state space and action space with complex transition dynamics, making it difficult to achieve steady-state. The noise and uncertainty in current neural network-based policies enable us to find the out-of-distribution primitives even though they are not optimal for primal tasks.
>
> ---
>
> Reference:
>
> [1] Eysenbach, B., Gupta, A., Ibarz, J., & Levine, S. (2018, September). Diversity is All You Need: Learning Skills without a Reward Function. In International Conference on Learning Representations.
>
> [2] Campos, V., Trott, A., Xiong, C., Socher, R., Giró-i-Nieto, X., & Torres, J. (2020, November). Explore, discover and learn: Unsupervised discovery of state-covering skills. In International Conference on Machine Learning (pp. 1317-1327). PMLR.
>
> [3] Laskin, M., Liu, H., Peng, X. B., Yarats, D., Rajeswaran, A., & Abbeel, P. (2022). CIC: Contrastive Intrinsic Control for Unsupervised Skill Discovery. arXiv preprint arXiv:2202.00161.

---

> ### Author Response · Authors · 2022-08-06
> **Any post-rebuttal comment?**
>
> Dear Reviewer,
>
> Please kindly let us know if you have any post-rebuttal feedback. We hope our rebuttal, especially the parkour demo, could address your concerns and look forwards to hearing back your optimistic acknowledgment of this submission.
>
> Best,
> Authors

---

> ### Author Response · Authors · 2022-08-09
> **Have the response addressed the reviewer's concerns?**
>
> Dear Reviewer,
>
> Thank you for raising a number of questions in the initial review. We prepared a detailed response below. Could you please provide some feedback and reassess our paper? If something is still unclear, we would be happy to provide alternative explanations or clarifications on any other part of the paper.
>
> Thanks,
> Authors

---

### Author Response · Authors · 2022-08-02
**Common Response**

We thank all reviewers for the insightful reviews. We revised our submission and we summarize the major updates as follows:

1. We trained a bipedal Cassie robot to move forward in IsaacGym and test it in a parkour environment, where it combines sidestepping, tiptoeing, crouching, back-flipping, and forward jumping to finish this task. The anonymous demo video is available at https://drive.google.com/file/d/1k182aDoS7SyddeO4Gu_ls_FHeXwkLBeh/view?usp=sharing. It helps validate the generalizability of the proposed method for fine-grained control.

2. We conducted a new quantitative experiment to investigate the accuracy of Policy Dissection enabled control. Concretely, we trained a SOTA goal-conditioned quadrupedal Animal-C robot in IsaacGym. It can track a heading command and will serve as the goal-condition baseline. For a fair comparison, we directly dissect this policy and identified the yaw velocity primitive. Recall that a positive (negative) activation will increase (decrease) corresponding kinematic attributes. Therefore, a PID controller that takes the heading error as input can be used to automatically determine the output of this primitive. Note that for the primitive activation control the explicit heading command is set to 0 in the whole experiment. This yaw tracking experiment shows that the primitive activation control is comparable to goal-conditioned control. The anonymous demo video with plotted tracking error curve is available at https://drive.google.com/file/d/1KK3Xc4y1hMsxcZB8byret3Fl7TXiwtZo/view?usp=sharing

---

### Meta-Review · Area_Chair_eiXN · 2022-08-25

**Recommendation:** Accept
**Confidence:** Certain

**Metareview:**

This paper proposes to dissect a trained policy, which finds the correspondence between the neuron activations and the motion primitives. This enables a human to control the agent to complete complex and diverse tasks even though only one simple task is trained. All reviewers agree that the proposed method is a creative solution to an important problem. And it may also have important applications in robotics. In addition, this is an important step towards understanding/interpreting neural network policies, which may inspire follow-up works in the future.

There are a few concerns and suggestions in the original reviews. Most of them are sufficiently addressed in the rebuttal and discussions. The new experimental results look impressive, and help to resolve a major concern about the coarseness of the human control that this paper enables. While the paper writing is still rough, the creative solution and the potential important applications compensate for the shortcomings. Thus, I recommend accepting this paper. Please revise the paper by incorporating reviewers' comments.

**Award:**

Yes

---

### Decision · Program_Chairs · 2022-09-14

Accept